# BENCHMARKING OPEN-ENDED SEGMENTATION

**Cristina González**[1,2,3]**, Santiago Rodríguez**[1]**, Kevis-Kokitsi Maninis**[4]**,**
**Jordi Pont-Tuset**[4]**, Pablo Arbeláez**[1]
[1]Universidad de los Andes [2]Helmholtz Munich [3]Technical University of Munich [4]Google DeepMind
`{ci.gonzalez10,s.rodriguezr2}@uniandes.edu.co`,
`{kmaninis,jponttuset}@google.com,pa.arbelaez@uniandes.edu.co`

## ABSTRACT

Open-ended segmentation requires models capable of generating free-form descriptions of previously unseen concepts and regions. Despite advancements in model development, current evaluation protocols for open-ended segmentation tasks fail to capture the true semantic accuracy of the generated descriptions. We empirically demonstrate that embedding-based similarity score mappings diverge significantly from human judgments. To address this issue, we introduce a novel mapping function that considers multiple lexical relationships between free-form outputs and test-vocabulary labels, yielding much closer alignment with human annotations. We integrate this mapping into a robust evaluation framework and re-benchmark previous state-of-the-art methods. Additionally, we present the first Multi-modal Large-Language Model trained with a contrastive objective to jointly align visual regions and textual descriptions, achieving new state-of-the-art results in open-ended panoptic segmentation.

## 1 INTRODUCTION

Recent advances in visual recognition have enabled the successful deployment of artificial intelligence in relatively controlled scenarios, under the requirement of high-quality and application-specific training data. However, the real world is dynamic, diverse, and unpredictable. To operate in such unconstrained environments, research has shifted toward visual recognition in open-ended settings (Radford et al., 2021; Jia et al., 2021; Zhang et al., 2023; Chen et al., 2023; You et al., 2023; Yuan et al., 2024). Unlike traditional visual recognition tasks (Deng et al., 2009; Lin et al., 2014; Kirillov et al., 2019), which rely on predefined and closed sets of semantic categories, open-ended settings require models to describe the semantics of visual entities by generating free-form language instead of choosing one label within a finite set of options.

Experimental frameworks for visual recognition incrementally introduce challenges along two axes: finer spatial localization and semantic generalization. In the former, the benchmarks have progressed from coarse labels at the image-level (Deng et al., 2009) to fine-grained, window- (Lin et al., 2014) or pixel-level (Lin et al., 2014; Zhou et al., 2017; Cordts et al., 2016; Gupta et al., 2019) annotations for a predefined set of categories. In terms of semantic generalization, early efforts (Deng et al., 2009; He et al., 2017; Cheng et al., 2022) operated in a strictly closed-set regime, training specialist models to recognize the concepts seen during training for a specific dataset only. Open-vocabulary frameworks (Li et al., 2022; Ghiasi et al., 2022; Xu et al., 2022) allow generalizing models to work on unseen categories by defining the vocabulary to operate on as an input to the model. Despite these advances, these benchmarks are still restricted to classifiers within a limited vocabulary, and evaluation is simplified to measuring accuracy against a "correct" answer within it.

Open-ended approaches (Radford et al., 2021; Jia et al., 2021; Zhang et al., 2023; Chen et al., 2023; You et al., 2023; Yuan et al., 2024) enable the segmentation of novel objects at test time without assigning them a predefined label, but instead generating free-form descriptions. These can differ in wording, granularity, or structure while still conveying the same concept. Multiple valid descriptions (e.g. "yellow dog," "golden retriever," or "a dog's tail") can refer to the same entity, and thus rigid label-matching metrics fall short in capturing semantic correctness. This one-to-many relationship between ground-truth concepts and acceptable linguistic expressions motivates the need for evaluation methods that go beyond exact string matching.

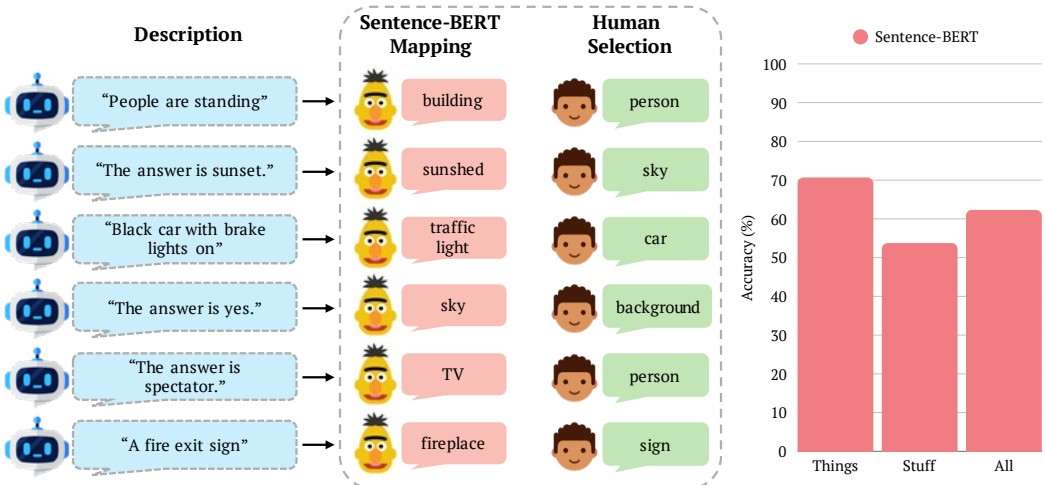

Figure 1: **Sentence-BERT mapping quality.** Examples of Sentence-BERT (Reimers & Gurevych, 2019) mappings to ADE20K (Zhou et al., 2017) and Cityscapes (Cordts et al., 2016) list of categories alongside the human-verified true category (left). We conducted a systematic user study to evaluate the quality of Sentence-BERT's mappings on the standard Cityscapes (Cordts et al., 2016) dataset that reveals an overall error with respect to human judgements of 37.7% (right).

Prior studies (Guo et al., 2024; Huang et al., 2024; You et al., 2023; Yuan et al., 2024) have repurposed metrics from related vision-language tasks to assess generalization in open-ended visual recognition, where free-form text outputs are typically scored with captioning metrics (Anderson et al., 2016; Banerjee & Lavie, 2005; Lin, 2004; Papineni et al., 2002; Vedantam et al., 2015). However, these metrics rely on region-description datasets that focus on "things" categories with window-level annotations drawn from semantically narrow sources, which limits their ability to measure true generalization towards unseen categories during training. Some works have turned to human user studies to rate output correctness, but these are labor intensive, expensive, and difficult to standardize.

Recently, researchers have established a standard evaluation framework for open-vocabulary segmentation, aiming to train on one segmentation dataset while testing on another, which includes entirely novel categories to assess semantic generalization directly. To evaluate open-ended segmentation approaches in this setup, prior work (Zhang et al., 2020) applied embedding-based similarity scores like Sentence-BERT (Reimers & Gurevych, 2019) to map free-form textual model outputs to the target taxonomy of categories in the test dataset. The metric maps a short description to the closest category in the pre-defined vocabulary in embedding space and then evaluates the performance of an open-ended model as if it had produced that category as output. Figure 1 (middle) shows examples of Sentence-BERT mappings to a target vocabulary alongside the human-verified true category. Notably, these mappings are often incorrect due to biases and limitations from the underlying embedding models, which directly affect the results of the evaluation of open-ended models in this setup. We conducted a systematic human verification to evaluate the quality of Sentence-BERT's mappings on the standard Cityscapes dataset that reveals an overall divergence of 37.7% from human judgments (Figure 1 (right)). These findings show that existing evaluation protocols fall short in capturing true semantic accuracy and open-ended generalization.

To address this limitation, this paper introduces a novel mapping function for the evaluation of open-ended segmentation that considers multiple lexical relationships among concepts, rather than relying on a single similarity score. Our proposed mapping function considered lists of valid words for each category in the test dataset, ordered by increasingly flexible lexical relationships (see Figure 2a). This approach better assigns free-form descriptions to the test vocabulary, resulting in significantly higher alignment with human annotations. We then integrate this lexical mapping into a comprehensive evaluation framework to re-evaluate several state-of-the-art open-vocabulary and Multi-modal Large Language Models (MLLMs), revealing systematic weaknesses in previous evaluation frameworks. Finally, we advance the field by introducing, to the best of our knowledge, the first MLLM trained with contrastive learning, which jointly aligns visual and textual representations for open-ended segmentation.

Our main contributions can be summarized as follows:

(1) We empirically demonstrate that current embedding-based similarity score metrics in existing open-ended segmentation benchmarks significantly diverge from human judgments, misestimating models' semantic accuracy.

(2) We establish a robust mapping from free-form outputs to the test vocabulary and an evaluation protocol for open-ended segmentation that considers multiple lexical relationships and aligns more closely with human judgment. We re-benchmark previous state-of-the-art methods to promote progress in open-ended segmentation.

(3) We introduce OPAL, the first MLLM trained with a contrastive learning objective to align visual regions and textual descriptions. This model achieves state-of-the-art results on open-ended panoptic segmentation.

To ensure the reproducibility of our results and promote further research in open-ended visual segmentation, we make all resources of this paper publicly available [1], including the implementation of the evaluation protocol with a re-benchmarking suite, as well as the source code for our method and the pretrained model.

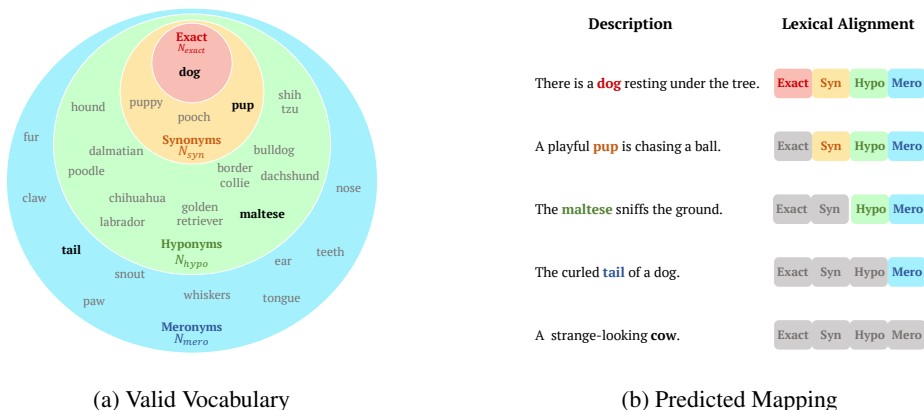

(a) Valid Vocabulary          (b) Predicted Mapping

Figure 2: **Lexical mapping.** (a) Example of the valid vocabulary for the target category "dog". (b) We illustrate how different descriptions are mapped by our proposed lexical mapping function to the "dog" semantic category at various lexical levels.

## 2 RELATED WORK

### 2.1 TOWARDS OPEN-ENDED SEGMENTATION

The field of visual recognition has progressed significantly over the years, starting with image recognition (Deng et al., 2009), where the task was to assign a single label to an entire image. The problem evolved into object detection (Lin et al., 2014), which added spatial localization by identifying bounding boxes around objects of interest. Advances then led to segmentation tasks such as semantic segmentation (Lin et al., 2014; Zhou et al., 2017; Cordts et al., 2016), which classified pixels into predefined categories, and instance segmentation (Lin et al., 2014; Gupta et al., 2019), which additionally differentiated between individual objects. A more recent development, panoptic segmentation, unifies both approaches to provide a complete understanding of a scene by distinguishing objects ("things") and amorphous regions ("stuff") (Forsyth et al., 1996; Adelson, 2001; Heitz & Koller, 2008; Kirillov et al., 2019). However, these advances have operated mainly within a closed-set paradigm, relying on predefined category sets during training and evaluation. While these methods have demonstrated outstanding performance on fixed category sets, real-world visual understanding demands capabilities beyond recognizing a limited set of predefined classes. Models trained under the closed-world paradigm tend to be dataset-specific and overly specialized, limiting their generalization ability to novel scenarios.

---

[1] https://github.com/BCV-Uniandes/open-ended_segmentation_benchmark

Open-vocabulary recognition (Li et al., 2022; Ghiasi et al., 2022; Xu et al., 2022) emerged as an important step toward more flexible visual understanding. The goal of this task is to extend classification beyond the restricted set of base classes by leveraging pretrained language models (Radford et al., 2021; Jia et al., 2021) to generalize across a broader vocabulary and recognize categories not explicitly seen during training at inference time. This task relies on fixed semantic labels during the evaluation, but allows the model to predict unseen categories by aligning visual features with language embeddings. Although open-vocabulary recognition expands the scope of categories a model can handle, it remains constrained by its dependence on predefined labels for evaluation. In contrast, open-ended segmentation (Radford et al., 2021; Jia et al., 2021; Zhang et al., 2023; Chen et al., 2023; You et al., 2023; Yuan et al., 2024; Reichard et al., 2025) requires models to generate natural language descriptions of visual regions without predefined labels, introducing a generative component to the task. This distinction makes evaluation significantly more challenging, as generated descriptions can vary widely in form while still being semantically accurate.

## 2.2 EVALUATION METRICS

In open-vocabulary settings, evaluation commonly relies on zero- or few-shot classification protocols that constrain model predictions to a fixed taxonomy. Evaluation strategies are often adapted from vision-language tasks to capture the semantic flexibility of model outputs. As a result, current evaluation approaches for open-vocabulary segmentation typically fall into three broad categories: captioning-based scores, embedding-based similarity measures, and human studies. While each provides valuable insights, all exhibit notable limitations that hinder their ability to assess the performance of open-ended segmentation models fully.

Captioning metrics such as BLEU (Papineni et al., 2002), METEOR (Banerjee & Lavie, 2005), and CIDEr (Vedantam et al., 2015) quantify n-gram overlap between generated and reference descriptions to evaluate the quality of automatically generated image captions. Additionally, SPICE (Anderson et al., 2016) incorporates scene-graph similarity for better semantic assessment. Still, these metrics were designed initially for constrained captioning tasks and often correlate poorly with human judgments in more complex or open-ended settings. Embedding-based metrics like Sentence-BERT (Zhang et al., 2020) and CLIPScore (Hessel et al., 2021) leverage pre-trained language models to assess similarity and are more robust to lexical variations by leveraging high-dimensional spaces to map from a description to a category. However, the underlying pre-trained models' biases, limitations, and embedding quality are inherited according to the semantic concepts present in their original training data. Human studies involve collecting evaluations from human annotators who assess model outputs' relevance, accuracy, or correctness through rating scales, preference judgments, or pairwise comparisons. This evaluation methodology serves as the gold standard in complex or ambiguous cases where automated metrics fall short, as they reflect real-world human judgment. However, this evaluation metric remains expensive, subjective, and difficult to scale or standardize across benchmarks.

Despite their strengths, existing automated evaluation metrics are largely designed for fixed-world scenarios, making them inadequate to capture the semantic complexity and variability of open-ended segmentation. This limitation underscores the need for new metrics designed for the unique challenges of open-ended settings. To address this gap, we introduce a new automated and standardized evaluation protocol designed for open-ended segmentation, significantly more aligned with human judgment than existing automated approaches.

## 2.3 OPEN-ENDED SEGMENTATION METHODS

Open-vocabulary segmentation methods have significantly advanced vision-language understanding by enabling models to recognize and segment arbitrary concepts at the pixel level, rather than being limited to a fixed set of categories. This progress is built on foundational works like CLIP (Radford et al., 2021) and ALIGN (Jia et al., 2021), which used contrastive learning on large-scale image–text datasets to achieve strong semantic alignment and generalization. Recent open-vocabulary segmentation methods (Xu et al., 2023; Dong et al., 2023; Liang et al., 2023) have extended these ideas from global image-level tasks to region- and pixel-level understanding, enabling models to align visual regions with diverse textual concepts and perform well on novel categories. Moreover, some state-of-the-art works (Rewatbowornwong et al., 2023; Ülger et al., 2025; Shin et al., 2024) have gone

beyond open-vocabulary segmentation and incorporated language models to automatically generate the input vocabulary solely from visual information, to finally perform mask classification.

More recently, MLLMs have proven highly effective thanks to their ability to process visual inputs alongside natural language instructions and generate free-form text, allowing them to predict novel or unseen categories. By incorporating both global visual features and localized region embeddings into prompts, these models can be queried about specific image regions. Early MLLM-based approaches used bounding boxes as input regions (Chen et al., 2023; Zhang et al., 2023). However, bounding boxes often include irrelevant background content, introducing noise that impairs recognition performance. To address this, recent works (Yuan et al., 2024; You et al., 2023; Shi et al., 2025) replaced bounding boxes with pixel-level masks, significantly improving accuracy by providing finer-grained inputs that yield more precise and context-aware predictions. However, while open-vocabulary segmentation has successfully leveraged contrastive learning to align regions with textual information (Xu et al., 2023; Dong et al., 2023; Liang et al., 2023) and achieved strong results, such concepts have been largely underexplored in open-ended segmentation, suggesting promising directions to refine training methodologies. In this work, we introduce, to the best of our knowledge, the first MLLM trained with a contrastive learning objective to align visual regions with textual descriptions.

## 3    LEXICAL ALIGNMENT METRICS

**Task Formulation:**    Open-ended segmentation aims at generating accurate semantic descriptions and segmenting visual regions within an image. Formally, given an input image $I \in \mathbb{R}^{H \times W \times 3}$, the goal is to produce binary masks $\{M_i\}_{i=1}^N$, where $M_i \in \{0, 1\}^{H \times W}$, and language descriptions $\{T_i\}_{i=1}^N$ according to the semantic information contained within each visual entity.

**Evaluation Metrics**    We follow established open-vocabulary evaluation protocols (Li et al., 2022; Ghiasi et al., 2022; Xu et al., 2022) to assess semantic generalization by training on one dataset and evaluating, without fine-tuning, on a distinct dataset whose vocabulary only partially overlaps with the training set. This protocol ensures that performance reflects true open-ended understanding rather than simply memorizing labels by introducing novel categories and altering label frequencies at test time.

Existing high-quality segmentation benchmarks (Cordts et al., 2016; Zhou et al., 2017), provide image regions with a unique ground-truth label within a closed-set vocabulary. However, since open-ended segmentation aims at generating free-form language descriptions, a mapping function is required to align these outputs with the target taxonomy of the testing benchmark. Previous approaches typically use embedding-based similarity scores (e.g., Sentence-BERT (Zhang et al., 2020)) to match each description to the category with the highest similarity. Nonetheless, this "forced choice" design requires every output to be assigned to one of the available labels, even when none adequately represent the described concept, leading to systematic misassignments. Moreover, this mapping function inherits the biases and limitations of the underlying embedding models, and its accuracy fluctuates with embedding quality rather than reflecting the model's true visual understanding.

To address the limitations of current approaches, we design a mapping function that relies on lexical relationships among nouns. Our lexical mapping function operates on a many-to-many basis, supporting three key characteristics that align with human judgment: (i) semantic accuracy: descriptions associated with each category must be semantically precise, preventing misclassification when no meaningful lexical relationship exists.; (ii) flexibility: each category can be related to multiple descriptions, recognizing that different sentences can accurately convey the same concept; and (iii) lexical proximity: a single description can have multiple lexical relationships with different categories on varying levels of lexical granularity, giving space to hierarchical taxonomies in the target vocabulary.

To meet the characteristics defined above, we start by considering a comprehensive list of nouns mined from a large-scale image-text pair dataset (Huang et al., 2023). This list of candidate nouns may contain compound nouns. We then use an LLM to identify which categories are semantically associated with each noun, using the prompt shown in Figure 6 of Appendix A. Finally, we compile a list of nouns that are lexically related to each category $c_i$ in the target vocabulary $\{C_i\}_{i=1}^N$. Our lexical mapping function considers multiple lexical relationships between the generated descriptions $\{T_i\}_{i=1}^N$ and potential semantic categories $\{C_i\}_{i=1}^N$. Thus, we propose organizing them in different

lexical levels, including exact matches, synonyms, hyponyms, and meronyms (see Figure 2a). Exact matches refer to descriptions that are matched directly by comparing the strings. Synonyms involve descriptions that convey the same or similar meaning using different words. Hyponyms refer to descriptions that indicate a more specific instance or subtype of the target category, while meronyms describe parts or components of the target category. These lexical levels are cumulative, meaning each level builds upon the relationships defined in the previous levels. This structure allows for a nuanced and hierarchical alignment of descriptions and categories.

Given a language description $T_i$ and a target vocabulary $\{C_i\}_{i=1}^{N}$, we define a function $f : T \to \{C_i\}_{i=1}^{N}$ that maps the output description to a list of all categories with which a lexical relationship exists, based on the specified lexical level. Figure 2 considers a target category "dog" and its associated valid vocabulary (Figure 2a) to illustrate how different descriptions are mapped to this semantic category at various lexical relationship levels (Figure 2b). Particularly, to process a predicted free-form description, we use automatic sentence positional tagging methods to extract the sentence's subject, identify its singular form, and define a lexical relationship between the description and a category if the extracted subject appears in the list of nouns associated with the category for each lexical level considered. Since standard recognition metrics typically require assigning a single category, the description is mapped to the ground-truth category if a lexical relationship with that category is found. Otherwise, the description is assigned to the category with the strongest available lexical relationship up to the given level. Suppose no lexical relationship can be established with any target category. In that case, the description is assigned to a background category, ensuring the output remains meaningful within the task context.

Using this approach, we obtain a mapping for a description at each lexical level, which allows us to calculate the desired recognition metric. We compute these metrics across all points along the semantic dimension to evaluate the model's overall performance. The results are plotted as a curve, with the x-axis representing the semantic dimension (i.e., lexical levels evenly spaced) and the y-axis showing the metric value. We name our evaluation protocol as the Lexical Alignment Curve (LAC). To provide a comprehensive measure of a model's performance, we compute the area under the curve that aggregates all levels of lexical alignment. This evaluation protocol is compatible with any standard recognition metric, making it broadly applicable across diverse open-ended segmentation tasks. Moreover, LAC enables diagnostic insights into how precise, generic, or ambiguous the model's descriptions are, based on its performance at each lexical level.

# 4   OPAL

We introduce the OPen ALignment method (OPAL), which enhances open-ended segmentation by introducing a contrastive training objective alongside the standard generative loss, building upon the Osprey (Yuan et al., 2024) architecture. This complementary learning objective enables the model to better aligned textual and fine-grained visual information in the embedding space which results in more robust generation capabilities. The core components remain unchanged: a CLIP-based (Radford et al., 2021) vision encoder extracts multi-scale visual features, which are projected into the language embedding space via a visual projector. A mask-aware visual extractor computes embeddings specific to segmented regions, enabling precise representations of visual entities. These embeddings are then processed by a LLaMA (Touvron et al., 2023) language model, fine-tuned with Low-Rank Adaptation (Hu et al., 2022) (LoRA), which generates detailed semantic descriptions conditioned on the visual context. We provide implementation details in Appendix B.2.

As shown in Figure 7 of Appendix B.1, the training process jointly optimizes two complementary loss functions, each requiring a different forward pass. The generative loss follows the standard approach, where the LLaMA model is prompted using visual and mask embeddings to generate a description. This learning objective encourages fluent and contextually relevant outputs. The contrastive loss aligns textual descriptions to regions in a joint embedding space. For this objective, only the textual description is included in the LLaMA prompt, and contrastive learning is performed between the last language embedding and the mask embedding. This function ensures tighter semantic alignment between the visual and textual modalities. We report the computational complexity overhead of the contrastive loss in Appendix B.3.

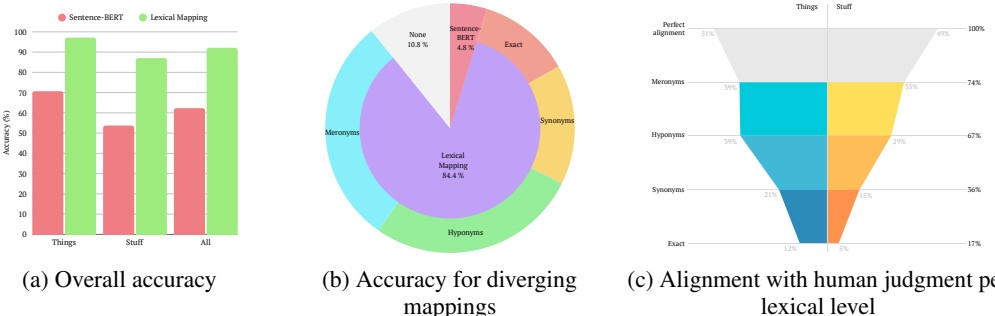

| (a) Overall accuracy | (b) Accuracy for diverging mappings | (c) Alignment with human judgment per lexical level |

Figure 3: **Human verification results.** (a) We report the accuracy of Sentence-BERT mapping vs our proposed Lexical mapping with respect to the human-selected category. Our lexical mapping function achieves a better alignment with human judgements. (b) Furthermore, we compute the accuracy with respect to the human label for the subset of descriptions where our mapping and Sentence-BERT mapping differs. (c) Finally, we evaluate our lexical mapping alignment with human selected categories when prioritizing ground-truth region labels.

## 5 EXPERIMENTS AND DISCUSSION

### 5.1 HUMAN VERIFICATION

To evaluate automated mapping functions, we conducted a two-stage human verification over 2800 region-level descriptions from the Cityscapes (Cordts et al., 2016) validation set, specifically targeting instances where Sentence-BERT (Reimers & Gurevych, 2019) and our mapping function disagreed at one or more lexical levels. The descriptions were generated by both Osprey-7B (Yuan et al., 2024) and OPAL to introduce linguistic variability. In the first stage, annotators assigned one of the Cityscapes semantic categories, or "None" if no valid match was found, to each region description. Disagreement was observed in only 2% of cases. These were then re-annotated independently in a second stage, resulting in a 100% agreement rate across all final annotations used for evaluation. This high inter-annotator agreement underscores the clarity and consistency among humans in performing the task. In Appendix C.1, we present additional details on the annotation interface (Figure 8) and show qualitative results (Figure 9).

We compute accuracy for Sentence-BERT mapping and our proposed lexical mapping function against human judgments. Particularly, for our mapping function we consider it a true positive if the human-selected category is between the valid vocabulary for any lexical level. As shown in Figure 3a, our method aligns with human annotations in over 90% of cases and outperforms Sentence-BERT by more than 20% across "things" and "stuff" categories. Sentence-BERT achieves only 60% accuracy overall and performs especially poorly on "stuff" categories, with correct mappings in just 50% of cases.

To gain more insights about disagreement between the two mapping functions, Figure 3b considers only the subset of descriptions where our mapping and Sentence-BERT differ. In these high-disagreement examples, our method recovers the human-assigned category 84.4% of the time, while Sentence-BERT succeeds in only 4.8%. Moreover, our mapping outperforms Sentence-BERT at the "Exact" semantic level, indicating that the latter fails to assign the correct category even when there is an exact string match between the category and the description's subject.

To evaluate whether the heuristic described in Section 3 for selecting a single category when a description lexically relates to multiple target categories introduces bias or artificially inflates the metric, we measured the accuracy of our mapping against human annotations at each lexical level. Additionally, we disaggregated these results into "things" and "stuff" categories. Figure 3c shows that alignment between our lexical mapping function and human annotations improves as the vocabulary flexibility increases. These results confirm that prioritizing ground-truth labels within the valid vocabulary preserves precision and maintains strong alignment with human understanding across all semantic levels. Notably, our mapping performs consistently well on both "things" and "stuff" categories, achieving 74% accuracy at the meronym level, highlighting robust correspondence with human annotations.

Table 1: **Open-ended segmentation results** for panoptic(PQ), semantic (mIoU), and instance (AP) segmentation tasks evaluated with our proposed evaluation protocol on the validation sets of ADE20K (Zhou et al., 2017) and Cityscapes (Cordts et al., 2016) datasets. Best results are shown in bold.

| Method | ADE20K (Zhou et al., 2017) | | | Cityscapes (Cordts et al., 2016) | | |
|---|---|---|---|---|---|---|
| | PQ | mIoU | AP | PQ | mIoU | AP |
| OVSeg (Liang et al., 2023) | 26.2 | 25.3 | 19.9 | 30.7 | 34.3 | 18.9 |
| MaskCLIP (Dong et al., 2023) | 27.1 | 22.9 | 19.2 | 22.4 | 22.6 | 11.7 |
| MasQCLIP (Xu et al., 2023) | 38.6 | 34.2 | 26.7 | 43.8 | 45.6 | 23.4 |
| Baseline | 42.9 | **39.8** | 28.5 | 46.9 | 53.7 | 28.8 |
| Shikra-7B (Chen et al., 2023) | 33.9 | 25.5 | 21.8 | 27.4 | 29.2 | 16.3 |
| Ferret-7B (You et al., 2023) | 42.3 | 31.2 | 31.3 | 34.3 | 36.9 | 26.0 |
| GPT4RoI-7B (Zhang et al., 2023) | 44.5 | 36.5 | 33.7 | 39.7 | 40.6 | 25.3 |
| Osprey-7B (Yuan et al., 2024) | $46.6 \pm 1.1$ | $36.9 \pm 1.1$ | $36.4 \pm 1.1$ | $50.2 \pm 1.0$ | $55.4 \pm 0.7$ | $31.3 \pm 0.6$ |
| OPAL (Ours) | $\mathbf{48.8 \pm 0.6}$ | $38.9 \pm 0.6$ | $\mathbf{38.4 \pm 0.7}$ | $\mathbf{52.8 \pm 0.6}$ | $\mathbf{56.1 \pm 0.4}$ | $\mathbf{31.9 \pm 0.6}$ |

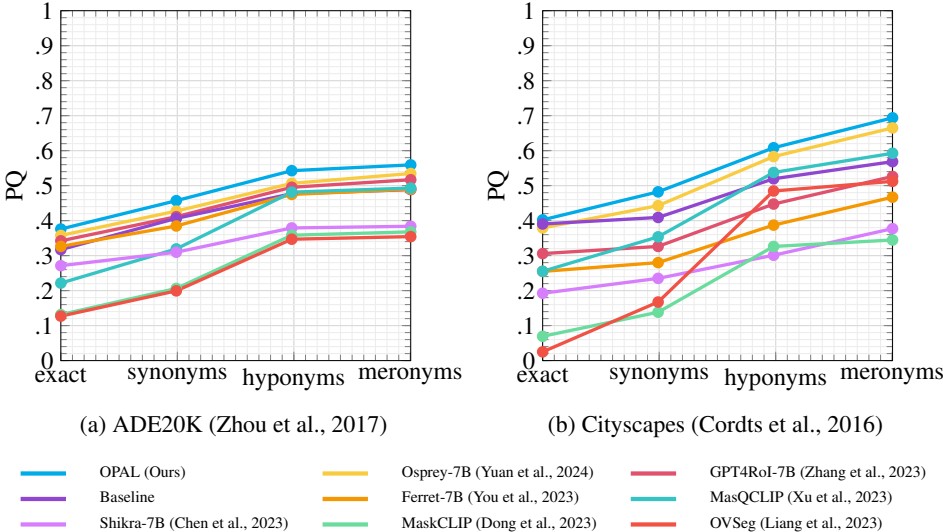

(a) ADE20K (Zhou et al., 2017)          (b) Cityscapes (Cordts et al., 2016)

Figure 4: **Lexical Alignment Curve.** We re-benchmark previous state-of-the-art methods using our proposed evaluation metric for panoptic segmentation in the validation set of (a) ADE20K (Zhou et al., 2017) and (b) Cityscapes (Cordts et al., 2016) datasets. OPAL outperforms both open-vocabulary and MLLM methods for all lexical levels across the two benchmark datasets.

## 5.2 MAIN RESULTS

To advance open-ended segmentation, we re-benchmark state-of-the-art models using their source code and pretrained weights with our proposed evaluation protocol. Our benchmarking covers models originally developed for open-vocabulary recognition, adapted to open-ended segmentation tasks using a comprehensive noun list as input vocabulary to approximate the open-ended target taxonomy. As a baseline, we introduce a simple approach leveraging MLLMs: first, an image is passed through an MLLM prompted to generate a comprehensive list of nouns describing the visual entities present in the scene (Figure 6 in Appendix C.2); then, these predicted image tags serve as the input vocabulary for an open-vocabulary recognition model. Specifically, we use MasQCLIP (Xu et al., 2023), as it is is the best-performing open-vocabulary method. Furthermore, we evaluate state-of-the-art MLLMs trained to process images and visual prompts in an open-ended setting. Finally, we compare the performance of our proposed method to validate our contributions.

Figures 4a and 4b present lexical alignment curves on the validation set of ADE20K (Zhou et al., 2017) (research-only, non-commercial) and Cityscapes (Cordts et al., 2016) (non-commercial, terms of use) datasets for open-ended panoptic segmentation. As expected, given our cumulative definition

of lexical levels, performance improves for all methods as semantic flexibility increases. When grouping methods based on whether they utilize MLLMs, we observe that non-MLLM methods exhibit a more pronounced performance jump from synonyms to hyponyms. This suggests that open-vocabulary models tend to select more specific semantic concepts than generation-based models. Moreover, we prove that our method achieves state-of-the-art results for all lexical levels. We present additional results in Figures 11 and 12 of Appendix C.2.

Table 1 depicts the overall lexical alignment metric for panoptic, semantic, and instance segmentation on the validation set of ADE20K and Cityscapes datasets. Our method outperforms all methods by at least 2 and 0.6 absolute points across all tasks on the ADE20K and Cityscapes datasets, respectively. We conducted inference experiments varying random seeds and generation hyperparameters for Osprey and our method. The box plots of the lexical alignment metric for the open-ended panoptic segmentation task (Figure 5) show that our model not only surpasses the previous state-of-the-art but also cuts output variance by nearly half. This demonstrates that incorporating a contrastive loss during training enhances model robustness.

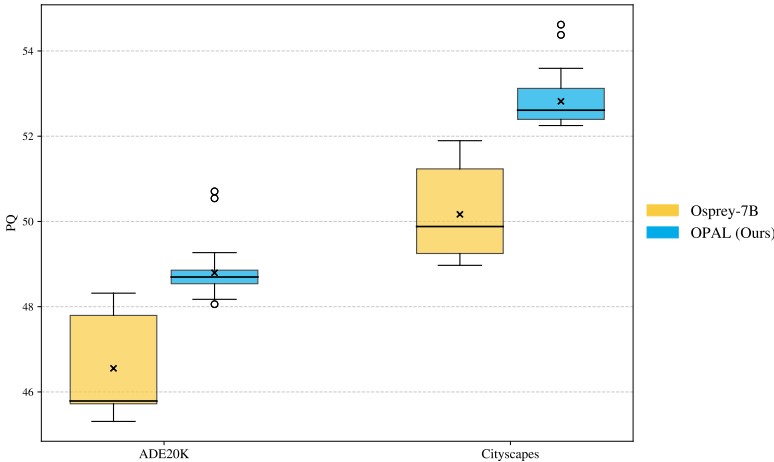

Figure 5: **Robustness analysis.** Box plots illustrating results on our proposed metric. OPAL outperforms Osprey and reduces output variance by nearly 50%, demonstrating enhanced robustness through contrastive learning training.

### 5.3 ABLATION STUDY

Table 2: **Lexical mapping impact across different LLMs.** We present open-ended segmentation results to compare the effect of using two different LLMs for the lexical mapping process. We include open-ended segmentation results for panoptic (PQ), semantic (mIoU), and instance (AP) segmentation tasks, evaluated using our proposed evaluation protocol on the validation sets of the ADE20K (Zhou et al., 2017) and Cityscapes (Cordts et al., 2016) datasets.

| Method | LLM | ADE20K | | | Cityscapes | | |
|---|---|---|---|---|---|---|---|
| | | PQ | mIoU | AP | PQ | mIoU | AP |
| Osprey (Yuan et al., 2024) | GPT-4 (Achiam et al., 2023) | 45.6 | 36.4 | 35.5 | 49.0 | 53.7 | 30.1 |
| | Gemini (Comanici et al., 2025) | 45.6 | 36.2 | 35.0 | 46.2 | 51.7 | 28.2 |
| OPAL (Ours) | GPT-4 (Achiam et al., 2023) | 47.9 | 37.9 | 37.6 | 52.4 | 55.3 | 32.7 |
| | Gemini (Comanici et al., 2025) | 48.0 | 37.7 | 37.1 | 48.3 | 51.7 | 30.2 |

**Impact of Lexical Mapping Across Different LLMs:** To further assess robustness to LLM-specific variability, we performed an ablation in which the lexical mapping process was repeated using two different LLMs. For all ablation experiments, we report the median results for OPAL and Osprey to account for computational complexity. To ensure fairness, we used identical prompts and set the temperature to 0.1 for all LLMs, minimizing stochasticity. Table 2 presents open-ended segmentation

results for panoptic (PQ), semantic (mIoU), and instance (AP) segmentation tasks, evaluated under our proposed evaluation protocol on the validation sets of ADE20K and Cityscapes, using GPT-4 and Gemini lexical mappings. The observed performance differences are negligible, and method ranking is preserved, providing additional empirical evidence that our framework is not overly sensitive to any particular LLM's biases.

**Impact of Lexical Coverage on Open-ended Segmentation:**  To address the effects of noun list size on mapping performance, we construct nested noun subsets containing 20, 40, 60, and 80% of the full list by randomly sampling the 20% subset and expanding it so that each larger subset contains the smaller ones. We ensure that all original dataset categories remain present in every subset. We then evaluate Osprey-7B and OPAL on the open-ended panoptic segmentation task using these progressively larger lists. As shown in Figure 13 in Appendix C.2 , our method outperforms Osprey-7B across all subset sizes, and LAC increases consistently for both models as the noun list grows, with a 6% gap between the full noun list and the 20% subset. This trend aligns with the nature of our metric: reducing the number of nouns narrows the set of matching candidates, increasing the likelihood that descriptions are mapped to the background category. Furthermore, the steady improvements with increasing noun set size highlight the value of comprehensive lexical coverage. These findings confirm that our mapping's nouns reliably represent the semantic concepts generated by an LLM in open-ended recognition tasks.

## 6   CONCLUSIONS

In this work, we introduce a novel evaluation protocol for open-ended segmentation that more accurately captures semantic alignment between generated free-form descriptions and target taxonomies. We empirically demonstrate that embedding-based similarity score mappings diverge significantly from human judgments through a systematic human verification. To address this issue, we design a mapping function that considers multiple lexical relationships, achieving closer alignment with human annotations. We integrate this mapping into a robust evaluation framework and re-benchmark previous state-of-the-art methods in open-ended segmentation tasks. Furthermore, we propose, to the best of our knowledge, the first MLLM trained with a contrastive objective to align visual regions and textual descriptions. OPAL achieves new state-of-the-art results on two standard benchmark datasets for open-ended panoptic segmentation.

## REPRODUCIBILITY STATEMENT

To ensure the reproducibility of our results and promote further research in open-ended segmentation, we provide details of the proposed metric in Section 3, as well as the prompt used to build the Lexical Mapping in Figure 6 of Appendix A. The proposed method is described in Section 4 and additional implementation details can be found in Appendix B.1. We also include additional results in Appendix C. Finally, all our source code and pretrained models are available at `https://github.com/BCV-Uniandes/open-ended_segmentation_benchmark`.

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

# A LEXICAL ALIGNMENT METRICS

```
system_message = [
{"role" : " system" ,
" content ": """Assistant is a chatbot trained to identify synonyms, hypernyms, holonyms, and semantic
relationships    of    nouns    according    to    a    predefined    list    of    categories.

Instructions:
- Consider the following list of semantic categories. Every group of words between "" and separated by comma is a
category (e.g. "plug, socket" is one category) : ["road", "sidewalk", "building", "wall", "fence", "pole", "traffic light",
"traffic sign", "vegetation", "terrain", "sky", "person", "rider", "car", "truck", "bus", "train", "motorcycle","bicycle"].
- The output should be exclusively in JSON format with the selected subsets of categories following this format:
{"synonyms": ["category_0", "category_1,…, "category_n"], "meronyms": ["category_0", "category_1,…,
"category_n"], "hyponyms": ["category_0", "category_1,…, "category_n"]}.
- DO NOT include categories outside the provided list.""""}
]

for noun in nouns:
    user_message = {"role" : " user", "content":{"type": "text", "text": f"Please identify synonyms, hyponyms, and
    meronyms of the noun {noun} from the predefined list of categories. "}}
    system_message.append(user_message )
```

Figure 6: **Lexical mapping generation prompt.** We use this prompt with ChatGPT/GPT-4 to construct our lexical mapping. Starting from a comprehensive list of nouns mined from a large-scale image-text dataset, we instruct the language model to identify which categories are semantically related to each noun. For every category in the target vocabulary, the prompt guides the extraction of lexically related terms. Specifically, we extract synonyms, hyponyms, and meronyms to support fine-grained semantic alignment between generated descriptions and the predefined category set.

# B OPAL

## B.1 METHOD OVERVIEW

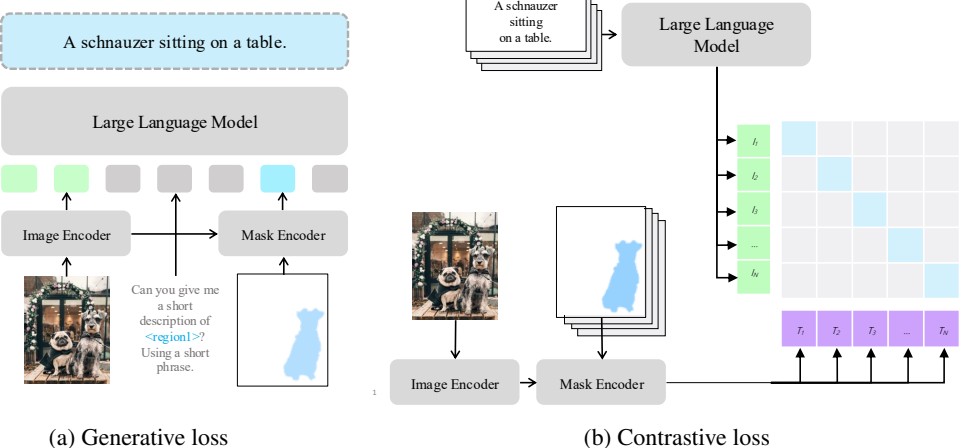

(a) Generative loss           (b) Contrastive loss

Figure 7: **OPAL: OPen ALignment method.** Our approach builds upon the Osprey architecture and introduces a dual-loss training strategy: (a) a generative loss, defined as next-token prediction for the output description given an image and a visual region; and (b) a contrastive loss, computed between region embeddings and the language embedding of the final token, which enforces tighter semantic alignment between visual and textual modalities.

## B.2 IMPLEMENTATION DETAILS

Our method is trained on Osprey-742K (Yuan et al., 2024) dataset (CC-BY-NC-4.0) to ensure fair comparison. We train our method on four NVIDIA A100 GPUs with 80GB memory. We maintain

the same setting used in Osprey for stage 1, which only trains the visual-language connector of the model between the pretrained visual encoder and LLM with image-text pair data. At the second stage, we decrease the learning rate to $5 \times 10^5$ with a batch size of 128 and train for four epochs. At the final stage, the learning rate is further reduced to $1 \times 10^5$ with a batch size of 64 for four epochs. The maximum length of sequence in LLM is set to 2048. We leverage the Deep-Speed framework for efficient large-scale model training. We use an AdamW (Loshchilov & Hutter, 2017) optimizer with standard parameters. To ensure training stability, we apply a 10% warmup, use a cosine annealing learning rate scheduler, and implement gradient clipping with a value of 1.0.

## B.3 COMPUTATIONAL COMPLEXITY OVERHEAD

The additional OPAL's computational overhead corresponds to only 1% of FLOPs during training, while maintaining the same computational complexity during inference. We consider our approach achieves a good trade-off between computational complexity and accuracy.

## C EXPERIMENTS

### C.1 HUMAN VERIFICATION

You will see a sentence describing a visual entity (such as an object, animal, body part, etc.). You must select from the dropdown menu the semantic category from the list that best describes the subject of the sentence. The list of possible semantic categories is as follows:

road, sidewalk, building, wall, fence, pole, traffic light, traffic sign, vegetation, terrain, river, sand, sea, snow, water, mountain, grass, dirt, rocks, sky, person, rider, car, truck, bus, train, motorcycle, bicycle, None of the above

The subject described may not always match the categories exactly, so you should choose the category that most closely corresponds to it. You may consider synonyms (e.g., if the subject is "automobile", select car), hyponyms (e.g., if the subject is "sedan", select car, since it is a type of car), or meronyms (e.g., if the subject is "wheel" and it is part of a bicycle, select bicycle) to identify the underlying concept. If the subject does not fit any of the 19 listed categories, select "None of the above".

Below are some examples of how categories should be assigned to different descriptions:

- "Red car stopped at a traffic light" – **car**
- "Sky with scattered clouds" – **sky**
- "Pedestrian standing next to a bus" – **person**
- "Graffiti on the wall of an old building" – **wall**
- "Person riding a motorcycle on the sidewalk" – **rider**
- "Corgi playing with a ball" - **None of the above**

Finally, keep the following considerations in mind:

1. If multiple objects are mentioned, select only the most relevant category.
2. If the description does not match any category, select "None of the above."
3. If you're unsure, ask yourself: What object or entity is the subject of the description?
4. Which semantic category best describes that object or entity?
5. Is any of the listed categories explicitly mentioned?

Figure 8: **Human verification instructions.** Instructions provided to annotators for evaluating region-level descriptions in the Cityscapes (Cordts et al., 2016) validation set. Annotators are asked to assign one of the official Cityscapes semantic categories to each description, or select "None" if the description does not refer to any target category. The study focuses on cases where Sentence-BERT (Reimers & Gurevych, 2019) and our mapping function disagree, and includes descriptions generated by Osprey-7B (Yuan et al., 2024) and our model.

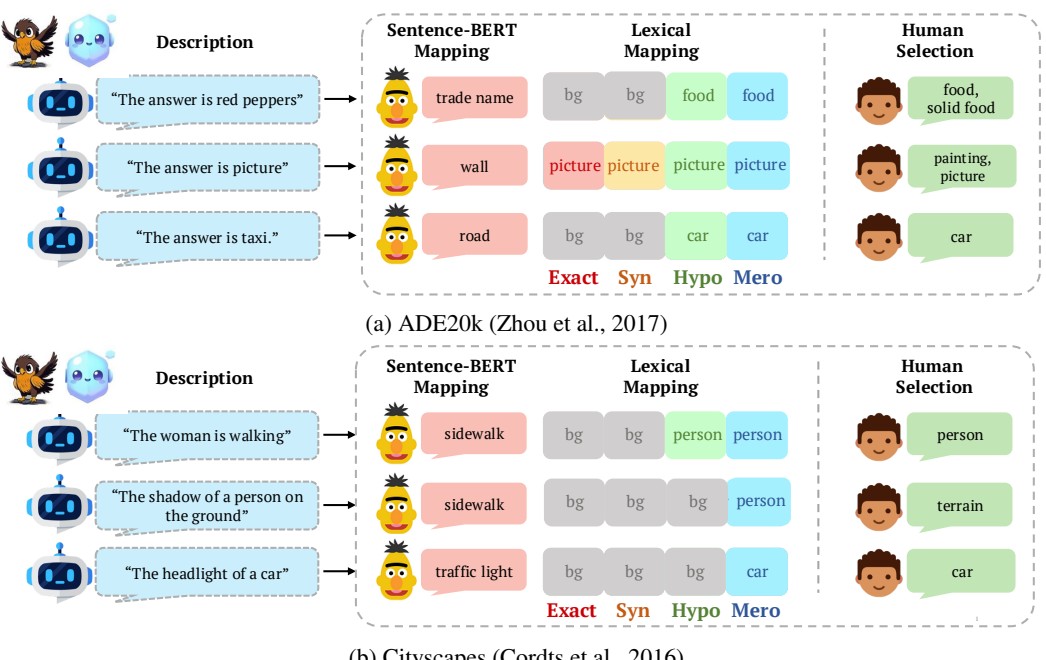

(a) ADE20k (Zhou et al., 2017)

(b) Cityscapes (Cordts et al., 2016)

Figure 9: **Sentence-BERT vs. our lexical mapping function.** Examples of Sentence-BERT (Reimers & Gurevych, 2019) and our Lexical Mapping to (a) ADE20k (Zhou et al., 2017) and (b) Cityscapes (Cordts et al., 2016) list of categories alongside the human-verified true category. The category *bg* stands for *background*.

## C.2 MAIN RESULTS

```
system_message = [
{"role" : " system" ,
"content" : """Assistant is a chatbot trained to describe images visually by providing a comprehensive list of all
semantic categories present in them.

Instructions:
-Scrutinize the entire image, including any areas that might first appear inconspicuous, to identify all things and stuff
categories.
-Exclude adjectives, verbs, or adverbs.
-The output should include categories for all objects and surfaces, even those that seem minor or part of the
background context.
-The output should be exclusively in JSON format with lowercase and singular categories following this format: {"
categories " : [" category_0 "," category_1 ",...," category_n "]}.
-Sort them by confidence.""" }
]

user_message = {"role" : " user","content":{"type": "text", "text": "Please provide a comprehensive list of all semantic
categories present in the image:"}}

for image in images:
    image_content = {"type": "image_url", "image_url": {"url": f"data: image /jpeg;base64,{base64_image}"}}
    system_message.append({"role" : " user", "content":[user_message, image_content]})
```

Figure 10: **Prompt for image tag extraction** We use this prompt to instruct ChatGPT/GPT-4 to generate a comprehensive list of nouns describing all visual entities present in the image. The resulting tags are used as the input vocabulary for the open-vocabulary recognition model MasQCLIP (Xu et al., 2023), which we use as our baseline.

When comparing the outputs of our model with those of Osprey-7B (Yuan et al., 2024) in Figure 11, we observe superior performance from our approach, especially on small objects. This suggests

that our model is more context-aware and better at understanding and extracting information from fine-grained image regions.

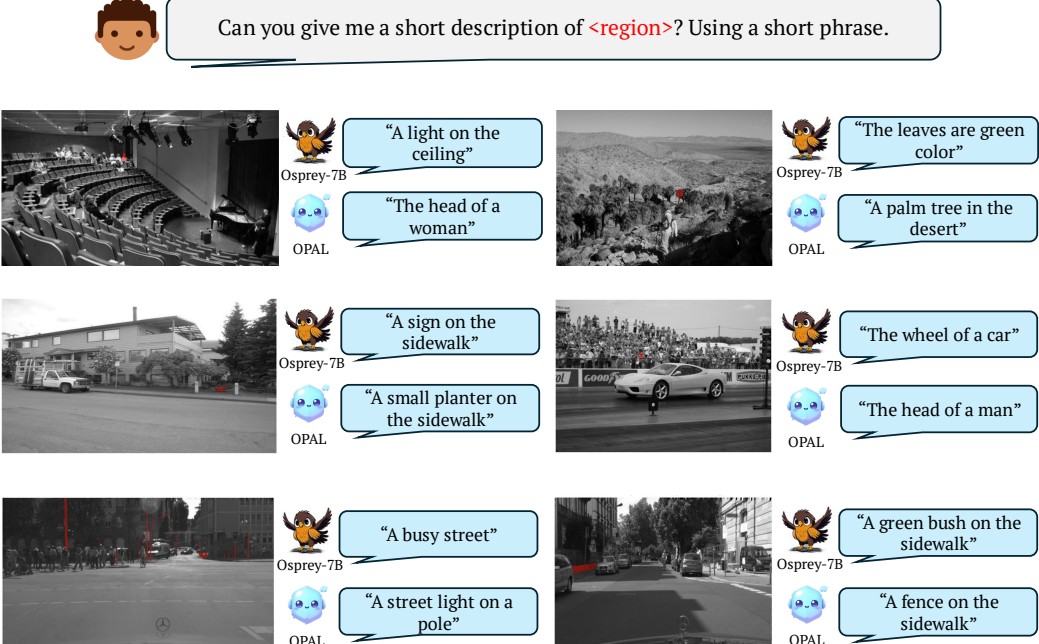

Figure 11: **OPAL's qualitative results.** Examples of OPAL output descriptions. We compare our outputs with those from Osprey-7B (Yuan et al., 2024), the second-best performing open-ended recognition model after ours.

Similar to the results presented in the main paper, Figure 12 shows a performance improvement in both instance and semantic segmentation as the level of lexical relation becomes more flexible. Our model consistently achieves state-of-the-art performance across all curves, demonstrating strong and reliable generalization across different metrics and datasets.

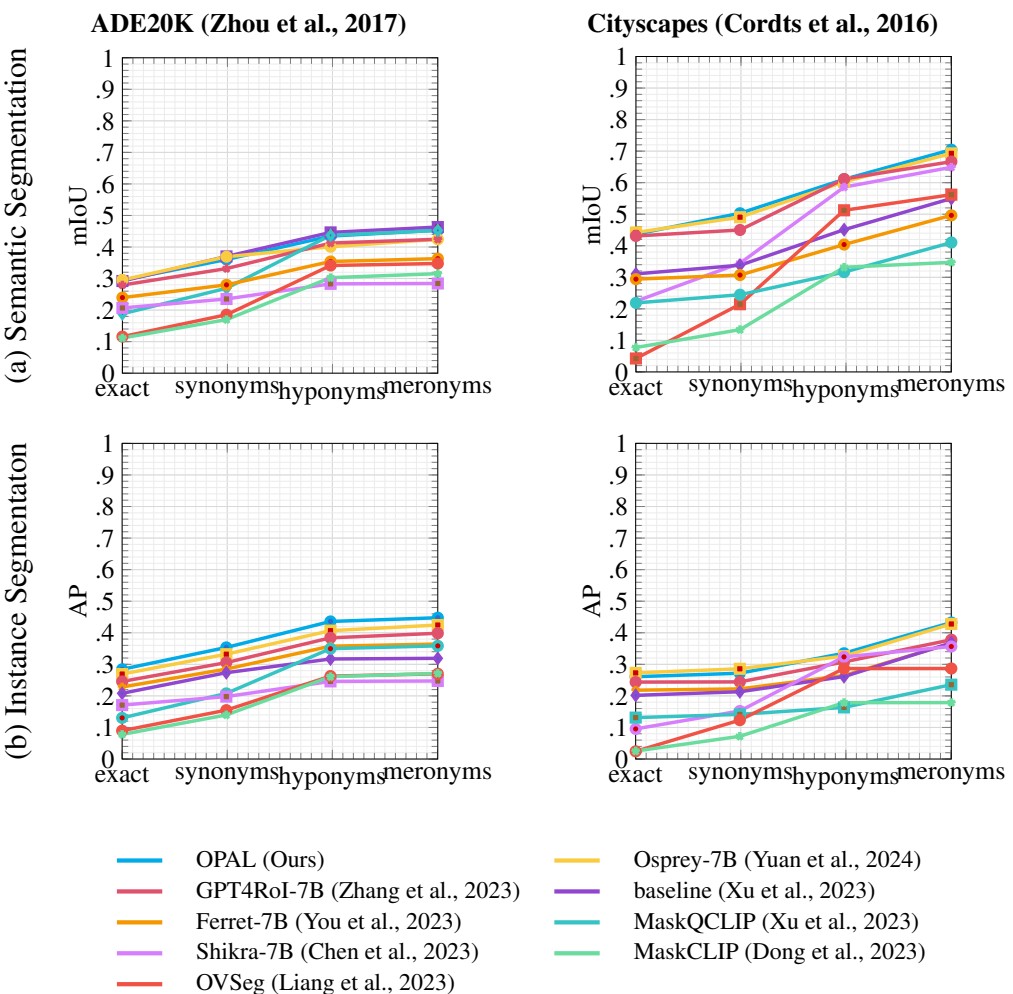

Figure 12: **Lexical Alignment Curve.** We report state-of-the-art methods results using the lexical alignment curve for instance and semantic segmentation in the validation sets of the ADE20K (Zhou et al., 2017) and Cityscapes (Cordts et al., 2016) datasets.

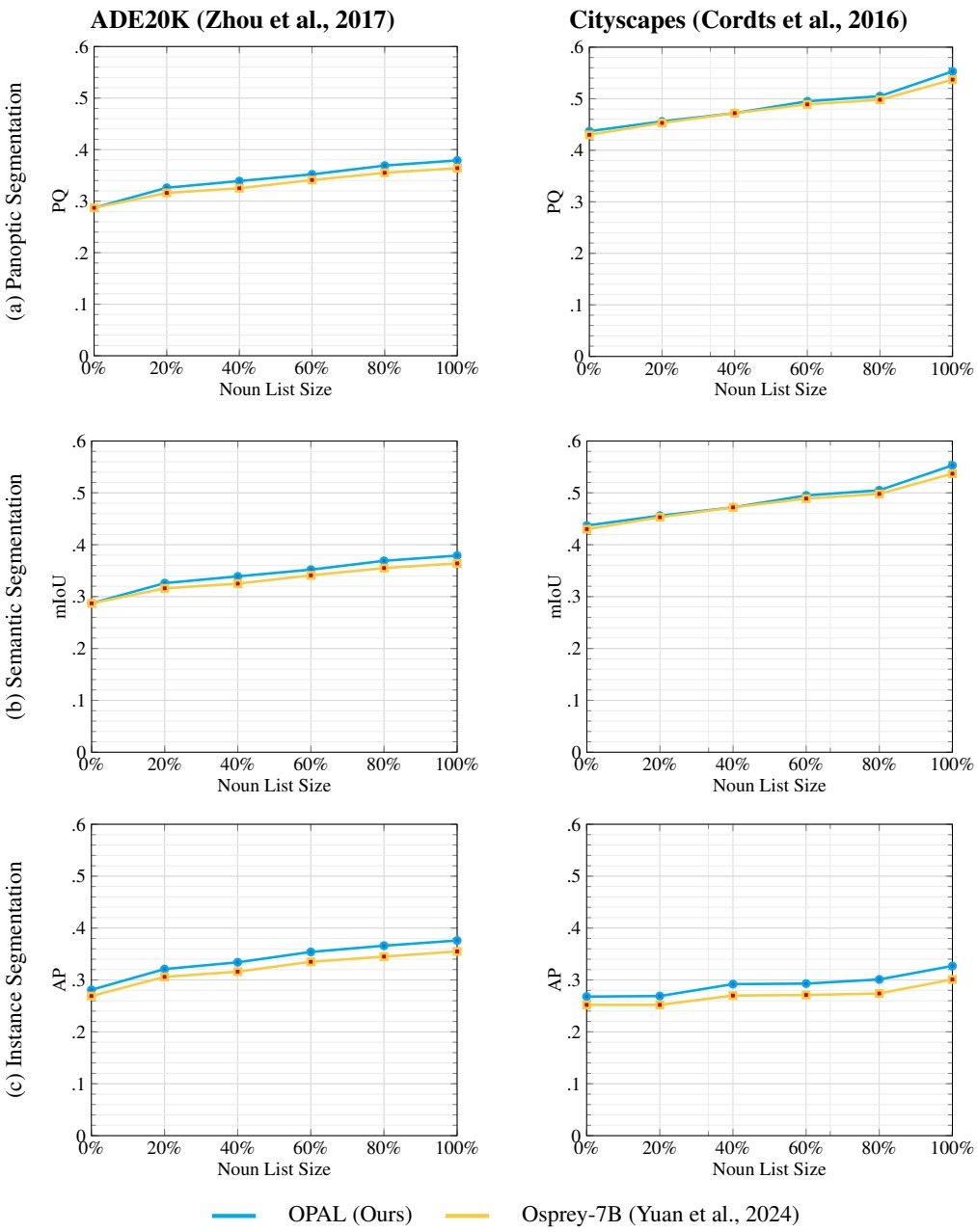

Figure 13: **Lexical coverage impact on open-ended segmentation.** We report lexical alignment curves across varying noun list sizes. We compare the performance of OPAL and Osprey-7B (Yuan et al., 2024) methods across panoptic, semantic, and instance segmentation in the validation sets of the ADE20K (Zhou et al., 2017) and Cityscapes (Cordts et al., 2016) datasets.

