# OpenReview forum: "Benchmarking Open-ended Segmentation"
_ICLR.cc/2026/Conference — ICLR 2026 Poster_

### Official Review · Reviewer_5AQi · 2025-10-30

**Soundness:** 3
**Presentation:** 3
**Contribution:** 3
**Rating:** 6
**Confidence:** 3

**Summary:**

This paper addresses flaws in open-ended segmentation (which needs models to generate free-form descriptions of unseen visual concepts, not select predefined labels) evaluation: it notes existing embedding-based metrics (e.g., Sentence-BERT) conflict with human judgments, and other metrics/studies are misaligned or impractical. It proposes a lexical relationship-based mapping (exact matches, synonyms, hyponyms, meronyms) integrated into the LAC framework, aligning better with human annotations. It also introduces OPAL, the first contrastive-trained MLLM for the task, achieving state-of-the-art results and better robustness.

**Strengths:**

It innovatively proposes a lexical relationship-aware mapping function integrated into the Lexical Alignment Curve framework and introduces OPAL, the first contrastive-trained MLLM for open-ended segmentation.

The LAC framework provides a reliable evaluation benchmark, OPAL sets a new SOTA.

It ensures rigor by comprehensively re-benchmarking SOTA models on ADE20K and Cityscapes via its proposed framework, and commits to releasing resources upon acceptance to guarantee reproducibility.

**Weaknesses:**

The paper’s lexical vocabulary lacks verification of its comprehensiveness (e.g., whether it covers low-frequency or niche concepts), which impacts the reliability of the evaluation results.

The paper has no ablation studies on critical lexical mapping designs (e.g., LLM selection basis, noun list filtering rules), undermining the reproducibility of the mapping.

The paper only validates the framework on ADE20K and Cityscapes, failing to include widely used datasets like COCO, limiting the generalization verification of its methods.

The paper does not report OPAL’s efficiency metrics such as inference latency and memory usage, making it hard to assess its practical deployment potential.

**Questions:**

Could you provide evidence of your lexical vocabulary’s coverage of rare/long-tail concepts?


could you add LLM-comparison ablation, release the noun list/LLM outputs, and quantify how noun list size/filtering affects mapping performance?

---

> ### Author Response · Authors · 2025-11-21
>
> We thank the reviewer 5Aqi for their feedback and positive rating of our work. In particular, the reviewer appreciated the novelty and robustness of our contrastively trained MLLM for open-ended segmentation together with our lexical relationship–aware mapping function, as well as the rigor of our approach in comprehensively re-benchmarking state-of-the-art models using the proposed framework. The reviewer also acknowledged our commitment to releasing all resources upon acceptance to ensure reproducibility. We also thank the reviewer for the thoughtful feedback. Below, we compile and directly address each raised question:
>
> * Could you provide evidence of your lexical vocabulary’s coverage of rare/long-tail concepts?
>
> To ensure compatibility and fairness with state-of-the-art methods (e.g., Osprey, Ferret, MasQCLIP), we adhere to the standard experimental setup for open-vocabulary segmentation evaluation. In this setup, to approximate an open scenario, models are evaluated solely on datasets not included in the training data, typically excluding the MS-COCO dataset used by all state-of-the-art methods. Since the LVIS dataset provides annotations only for MS-COCO training images and the COCO-Stuff dataset is included in the training data, they do not align with this experimental setup and would not allow us to benchmark the open-ended segmentation task.

---

> ### Author Response · Authors · 2025-11-21
>
> * Could you add LLM-comparison ablation, release the noun list/LLM outputs, and quantify how noun list size/filtering affects mapping performance?
>
> **Impact of Lexical Mapping Across Different LLMs:**
> To further assess robustness to LLM-specific variability, we performed an ablation in which the lexical mapping process was repeated using two different LLMs. For all ablation experiments, we report the median results for OPAL and Osprey to account for computational complexity. To ensure fairness, we used identical prompts and set the temperature to 0.1 for all LLMs, minimizing stochasticity. Table 2. presents open-ended segmentation results for panoptic (PQ), semantic (mIoU), and instance (AP) segmentation tasks, evaluated under our proposed evaluation protocol on the validation sets of ADE20K and Cityscapes, using GPT-4 and Gemini lexical mappings. The observed performance differences are negligible, and method ranking is preserved, providing additional empirical evidence that our framework is not overly sensitive to any particular LLM’s biases.
>
> Table 2. **Lexical mapping impact across different LLMs**. We present open-ended segmentation results to compare the effect of using two different LLMs for the lexical mapping process. We include open-ended segmentation results for panoptic (PQ), semantic (mIoU), and instance (AP) segmentation tasks, evaluated using our proposed evaluation protocol on the validation sets of the ADE20K and Cityscapes datasets.
>
> | |    |            |      **ADE20k**    |       |           |   **Cityscapes**    |       |
> |--------|--------|----------------------|-------|-------|----------------------|-------|-------|
> |   **Method**     |  **LLM**      | PQ                   | mIoU  | AP    | PQ                   | mIoU  | AP    |
> | Opal   | GPT-4  | 47.9                 | 37.9  | 37.6  | 52.4                 | 55.3  | 32.7  |
> |        | Gemini | 48.0                 | 37.7  | 37.1  | 48.3                 | 51.7  | 30.2  |
> | Osprey | GPT-4  | 45.6                 | 36.4  | 35.5  | 49.0                 | 53.7  | 30.1  |
> |        | Gemini | 45.6                 | 36.2  | 35.0  | 46.2                 | 51.7  | 28.2  |
>
> **Impact of Lexical Coverage on Open-ended Segmentation:**
> To address the effects of noun list size on mapping performance, we construct nested noun subsets containing 20, 40, 60, and 80\% of the full list by randomly sampling the 20\% subset and expanding it so that each larger subset contains the smaller ones. We ensure that all original dataset categories remain present in every subset. We then evaluate Osprey-7B and OPAL on the open-ended panoptic segmentation task using these progressively larger lists. As shown in Table 3., our method outperforms Osprey-7B across all subset sizes, and LAC increases consistently for both models as the noun list grows, with a 6\% gap between the full noun list and the 20\% subset. This trend aligns with the nature of our metric: reducing the number of nouns narrows the set of matching candidates, increasing the likelihood that descriptions are mapped to the background category. Furthermore, the steady improvements with increasing noun set size highlight the value of comprehensive lexical coverage. These findings confirm that our mapping’s nouns reliably represent the semantic concepts generated by an LLM in open-ended recognition tasks.
>
> Table 3. **Lexical coverage impact on open-ended segmentation**. We report lexical alignment curves across varying noun list sizes. We compare the performance of OPAL and Osprey-7B methods across panoptic, semantic, and instance segmentation on the validation sets of the ADE20K and Cityscapes datasets.
>
> |                     |                | **Cityscapes** |        |            | **ADE20K** |            |
> |---------------------|----------------|------------|------------|------------|------------|------------|
> |  **Noun Percentage (%)**| PQ             | mIoU       | AP         | PQ         | mIoU       | AP         |
> | 100                 | 45.6           | 36.4       | 35.5       | 47.9       | 37.9       | 37.6       |
> | 80                  | 44.4           | 35.5       | 34.5       | 46.5       | 36.9       | 36.6       |
> | 60                  | 42.9           | 34.1       | 33.5       | 44.9       | 35.2       | 35.4       |
> | 40                  | 40.8           | 32.5       | 31.6       | 43.1       | 33.9       | 33.4       |
> | 20                  | 39.6           | 31.6       | 30.6       | 41.8       | 32.6       | 32.1       |
>
> * The paper does not report OPAL’s efficiency metrics.
>
> The additional computational overhead of OPAL over Osprey corresponds to only 1% of FLOPs during training, while maintaining the same computational complexity during inference. We consider our approach achieves a good trade-off between computational complexity and accuracy. This analysis is in Appendix B.3 of the submitted paper.

---

> > ### Comment · Reviewer_5AQi · 2025-11-27
> >
> > Thank you for addressing my concerns. I believe the paper is worthy of acceptance to ICLR.

---

> > > ### Author Response · Authors · 2025-11-27
> > >
> > > Thank you for carefully reviewing our rebuttal. We are pleased that our responses addressed your concerns and appreciate your positive evaluation. We are grateful for all your efforts in helping us improve our submission. We have incorporated all suggested revisions into the updated version of the paper.

---

### Official Review · Reviewer_BTMF · 2025-11-01

**Soundness:** 2
**Presentation:** 2
**Contribution:** 3
**Rating:** 6
**Confidence:** 3

**Summary:**

This paper revisits the problem of evaluation of open-ended segmentation models.
The authors propose:
1. a lexical alignment–based evaluation protocol (LAC) that integrates hierarchical word relations—Exact, Synonym, Hyponym, Meronym—for mapping free-form text to target categories.
2. OPAL, a multi-modal large language model trained with a dual generative–contrastive objective to better align visual regions and textual outputs.

**Strengths:**

- The proposed lexical mapping and curve alignment method are novel. The paradigm provides a systematic and interpretable way to evaluate free-form segmentation outputs, filling an important methodological gap.
- The method is aligned with human verification (90% with human judgement).

**Weaknesses:**

- Dependence on external LLMs (e.g., GPT-4) for lexical relation extraction may reduce reproducibility and introduce hidden biases due to model updates.
- The metric maybe be hacked by the inclusion of hyponyms and meronyms.
- The 7B OPAL, as a tool in metric, is not convinent to deploy (specific enviroment). It may cause mismatched aligning and unfair comparison.

**Questions:**

- Is this method, maybe be used not only for open-ended segmentation? what about open-ended detection?
- What about contextual semantics or polysemy? (e.g., “apple” as fruit vs. company). How does the system handle cases where semantically distinct words share lexical proximity (e.g., “truck” → “car”)?
- How stable are the GPT-4–derived lexical relations across different LLMs, prompts, or randomness?

---

> ### Author Response · Authors · 2025-11-21
>
> We appreciate the reviewer’s positive assessment of the novelty of our work, including our proposed evaluation protocol and multi-modal large language model. We thank the reviewer for recognizing how our method addresses a key methodological gap by providing a systematic and interpretable way to evaluate free-form text descriptions, as well as the human verification we conducted. We also thank the reviewer for the thoughtful feedback. Below, we compile and directly address each raised question:
>
> * Is this method, maybe be used not only for open-ended segmentation? what about open-ended detection?
>
> Our work focuses on open-ended segmentation, since it is the most spatially fine-grained and challenging problem of visual recognition. Because our method operates on input masks, segmentation is the natural setting for evaluating its performance. However, the same framework can be adapted to open-ended detection. Bounding boxes can be derived from segmentation masks, enabling our recognition component to be applied directly to the detection task. In this sense, our approach is not limited to segmentation and can be extended to coarser formulations such as open-ended detection.
>
> * What about contextual semantics or polysemy? (e.g., “apple” as fruit vs. company). How does the system handle cases where semantically distinct words share lexical proximity (e.g., “truck”→ “car”)?
>
> Our approach can account for contextual semantics and polysemy by leveraging LLMs' broad world knowledge and generalization capabilities. For instance, if the subject of the output description has “apple” as a subject, we can consider it a hyponym of both fruit and company semantic categories. Since standard recognition metrics typically require assigning a single category, the description is mapped to the ground-truth category if a lexical relationship with that category is found. Otherwise, the description is assigned randomly to a category within the strongest available lexical relationship up to the given level.
>
> Moreover, our lexical mapping function is carefully designed to ensure the output remains meaningful within the task context. Therefore, we propose organizing nouns into distinct lexical levels: exact matches, synonyms, hyponyms, and meronyms. Exact matches refer to descriptions that are matched directly by comparing the strings. Synonyms involve descriptions that convey the same or similar meaning using different words. Hyponyms refer to descriptions that indicate a more specific instance or subtype of the target category, while meronyms describe parts or components of the target category. These lexical levels do not account for semantically distinct words that are lexically proximate, such as truck and car, since they are not synonyms, hyponyms, or meronyms.
>
> * How stable are the GPT-4–derived lexical relations across different LLMs, prompts, or randomness?
>
> **Impact of Lexical Mapping Across Different LLMs**:
> To further assess robustness to LLM-specific variability, we performed an ablation in which the lexical mapping process was repeated using two different LLMs. For all ablation experiments, we report the median results for OPAL and Osprey to account for computational complexity. To ensure fairness, we used identical prompts and set the temperature to 0.1 for all LLMs, minimizing stochasticity. Table 2. presents open-ended segmentation results for panoptic (PQ), semantic (mIoU), and instance (AP) segmentation tasks, evaluated under our proposed evaluation protocol on the validation sets of ADE20K and Cityscapes, using GPT-4 and Gemini lexical mappings. The observed performance differences are negligible, and method ranking is preserved, providing additional empirical evidence that our framework is not overly sensitive to any particular LLM’s biases.
>
> Table 2. **Lexical mapping impact across different LLMs**. We present open-ended segmentation results to compare the effect of using two different LLMs for the lexical mapping process. We include open-ended segmentation results for panoptic (PQ), semantic (mIoU), and instance (AP) segmentation tasks, evaluated using our proposed evaluation protocol on the validation sets of the ADE20K and Cityscapes datasets.
>
> | |    |            |      **ADE20k**    |       |           |   **Cityscapes**    |       |
> |--------|--------|----------------------|-------|-------|----------------------|-------|-------|
> |   **Method**     |  **LLM**      | PQ                   | mIoU  | AP    | PQ                   | mIoU  | AP    |
> | Opal   | GPT-4  | 47.9                 | 37.9  | 37.6  | 52.4                 | 55.3  | 32.7  |
> |        | Gemini | 48.0                 | 37.7  | 37.1  | 48.3                 | 51.7  | 30.2  |
> | Osprey | GPT-4  | 45.6                 | 36.4  | 35.5  | 49.0                 | 53.7  | 30.1  |
> |        | Gemini | 45.6                 | 36.2  | 35.0  | 46.2                 | 51.7  | 28.2  |

---

> ### Comment · Reviewer_BTMF · 2025-11-23
>
> Thanks for the rapid response.
>
> The authors seems miss the `Weaknesses` comment. They only answer the questions.

---

> > ### Author Response · Authors · 2025-11-24
> >
> > We thank the reviewer for raising these important points. We apologize for not adequately addressing them in our previous response. We now address each comment below:
> >
> > * Dependence on external LLMs (e.g., GPT-4) for lexical relation extraction may reduce reproducibility and introduce hidden biases due to model updates.
> >
> > To address the current bottleneck in evaluation metrics for open-ended visual segmentation, our framework leverages the broad world knowledge and generalization capabilities of LLMs while maintaining an interpretable evaluation pipeline. We agree with the reviewer that, since our lexical mapping relies on LLMs to extract synonyms, hyponyms, and meronyms, there may still be biases or inconsistencies associated with the LLM. However, our human verification results and ablation study confirm that the lexical relationships captured by the mapping align well with human judgment, and our framework is not overly sensitive to any particular LLM’s biases. Our empirical results suggest that our design choice of leveraging lexical relationships to make the evaluation more interpretable mitigates biases introduced by relying solely on pretrained models.
> >
> > Moreover, as mentioned in our “Reproducibility Statement,” we are committed to making all resources from this paper publicly available upon acceptance. To further promote transparency and fair evaluation, we provide both the evaluation protocol and a comprehensive re-benchmarking suite in the Supplementary Material. This includes the pre-computed lexical mapping function used to re-benchmark all methods, enabling the community to reproduce our results and fairly compare new approaches for open-ended segmentation.
> >
> > * The metric maybe be hacked by the inclusion of hyponyms and meronyms.
> >
> > To rigorously address concerns about potential bias from broader lexical relationships, we evaluated alignment with human judgment at each lexical level. As shown in Figure 3c of the main paper, alignment between our lexical mapping function and human annotations consistently improves as lexical coverage expands. These results provide strong evidence that our approach remains robust and is not biased toward artificially inflating the metric by including hyponyms and meronyms. This validation further highlights the reliability and fairness of our evaluation framework.
> >
> > Moreover, since the final LAC score is computed as the area under the curve across all lexical levels, the metric inherently favors matches at the stronger lexical levels (i.e., exact matches). Models that predict strong lexical relationships with the ground-truth category naturally accumulate performance at subsequent levels, resulting in a higher overall LAC score (as illustrated in Figure 2). Conversely, models whose predictions only align at broader levels (such as hyponyms or meronyms) suffer a substantial performance drop, since their scores at earlier levels remain very low and thus contribute little to the overall area under the curve. Therefore, our proposed metric is inherently robust against relying solely on broader lexical relationships since they cannot compensate for missing stronger lexical relationships with ground-truth labels.
> >
> > * The 7B OPAL, as a tool in metric, is not convinent to deploy (specific enviroment). It may cause mismatched aligning and unfair comparison.
> >
> > OPAL is our proposed MLLM, trained with dual generative and contrastive learning objectives for open-ended segmentation, and we benchmark it against previous state-of-the-art methods. Importantly, our evaluation protocol is entirely independent of OPAL itself and is implemented using the widely adopted detectron2 framework. By decoupling the evaluation protocol from the model and leveraging a standard implementation, we ensure that our work not only enables fair comparisons but also facilitates future research and reproducibility in open-ended segmentation.

---

> > > ### Comment · Reviewer_BTMF · 2025-11-25
> > >
> > > Thanks for your response, I do not have more questions.

---

> > > > ### Author Response · Authors · 2025-11-27
> > > >
> > > > Thank you for your response and positive rating of our work. We appreciate your thoughtful feedback and are happy that our rebuttal addressed your questions. Your comments were helpful in improving our submission, and we have incorporated all suggested revisions in the updated version of the paper.

---

### Official Review · Reviewer_zkjh · 2025-11-01

**Soundness:** 3
**Presentation:** 3
**Contribution:** 3
**Rating:** 6
**Confidence:** 4

**Summary:**

This paper addresses the challenge of evaluating open-ended segmentation, where models generate free-form textual descriptions instead of choosing from a fixed label set. The authors demonstrate that existing embedding-based evaluation methods (e.g., Sentence-BERT) poorly align with human judgment and propose a new Lexical Alignment Curve (LAC) metric that accounts for multiple lexical relationships—such as exact matches, synonyms, hyponyms, and meronyms—to better capture semantic correctness. They further introduce OPAL, the first multimodal large language model trained with a contrastive objective to jointly align visual regions and textual descriptions. Extensive experiments and human verification studies show that the proposed evaluation framework achieves stronger alignment with human perception and that OPAL sets new state-of-the-art results on open-ended segmentation benchmarks.

**Strengths:**

**Originality:** The paper tackles a highly relevant and underexplored problem—how to objectively evaluate open-ended segmentation models where outputs are free-form text rather than fixed labels. The proposed Lexical Alignment Curve (LAC) introduces a novel evaluation perspective that bridges semantic granularity levels (exact, synonym, hyponym, meronym). This is a creative and conceptually elegant formulation that advances beyond traditional embedding-based similarity scoring.

**Quality:** The methodology is systematic and carefully designed, combining empirical human studies, metric development, and benchmarking experiments.

**Clarity:** The paper is well-structured and logically presented—the motivation is clearly articulated, the limitations of existing methods are explained with concrete examples, and the proposed framework is illustrated with intuitive figures.

**Significance:** The work addresses a critical bottleneck in open-ended visual understanding: the lack of accurate, scalable evaluation metrics that align with human perception. The introduction of OPAL demonstrates practical impact by achieving measurable improvements on standard datasets, indicating both theoretical and empirical significance.

**Weaknesses:**

- The proposed lexical mapping relies heavily on automatic subject extraction to identify the key semantic concept in a sentence. However, this assumption breaks down for many common sentence structures where the grammatical subject is not the true visual referent (e.g., “A man is walking a dog,” “This is a picture of a dog,” or “On the grass, there is a dog”). Moreover, the method does not account for contextual modifiers such as adjectives or compound nouns, which can drastically alter meaning (e.g., “toy dog,” “hot dog,” “drawing of a dog”). As a result, the current approach may conflate semantically distinct entities or misidentify the referent.
- The lexical mapping depends on lexical resources (synonyms, hyponyms, meronyms) extracted using large language models and linguistic heuristics. However, this approach may **also** inherit biases and incompleteness from the LLM itself.
- While the proposed Lexical Alignment Curve framework effectively addresses the shortcomings of embedding-based mappings, its evaluation scope is relatively narrow. The analysis focuses primarily on Cityscapes and ADE20K, which, although standard, may not fully represent the diversity of open-ended segmentation scenarios (e.g., fine-grained or domain-specific datasets such as LVIS or COCO-Stuff).
- Although the paper is titled “Benchmarking Open-Ended Segmentation,” the experimental setup assumes that masks are provided as input, meaning that the model does not perform mask generation or spatial localization itself. Consequently, the task formulation aligns more closely with open-ended region or object recognition, rather than full open-ended segmentation.
- Several closely related works [1,2,3] are missing from the discussion.

[1] Shin, Heeseong, Chaehyun Kim, Sunghwan Hong, Seokju Cho, Anurag Arnab, Paul Hongsuck Seo, and Seungryong Kim. "Towards open-vocabulary semantic segmentation without semantic labels." Advances in Neural Information Processing Systems 37 (2024): 9153-9177.

[2] Ülger, Osman, Maksymilian Kulicki, Yuki Asano, and Martin R. Oswald. "Auto-vocabulary semantic segmentation." In Proceedings of the IEEE/CVF International Conference on Computer Vision, pp. 24266-24275. 2025.

[3] Rewatbowornwong, Pitchaporn, Nattanat Chatthee, Ekapol Chuangsuwanich, and Supasorn Suwajanakorn. "Zero-guidance segmentation using zero segment labels." In Proceedings of the IEEE/CVF International Conference on Computer Vision, pp. 1162-1172. 2023.

**Questions:**

- How does the proposed lexical mapping handle cases where the grammatical subject does not correspond to the true visual referent (e.g., “A man is walking a dog”)?
- Does the method explicitly account for modifiers such as adjectives or compound nouns (e.g., “toy dog,” “hot dog,” “drawing of a dog”)?
- Since the lexical mapping depends on large language models for synonym, hyponym, and meronym extraction, how do the authors address potential biases or inconsistencies in these LLM-generated resources?
- Could the authors discuss how their framework might generalize to fine-grained or domain-specific datasets such as LVIS or COCO-Stuff?
- Given that the experiments assume pre-defined masks as input, does the proposed setup still align with the notion of “open-ended segmentation”?

---

> ### Author Response · Authors · 2025-11-21
>
> We thank reviewer zkjh for their valuable insights and interest in our work, particularly for recognizing the novelty and creativity of our proposed evaluation framework, which offers a more accurate assessment of semantic alignment by accounting for multiple lexical relationships. We further appreciate the reviewer highlighting the quality of our systematic methodology, including human verifications, and the re-benchmarking of prior models under the new evaluation framework. Finally, we thank the reviewer for recognizing the clarity of our presentation, the significance of our evaluation framework, and OPAL's strong performance on open-ended recognition tasks. We appreciate the reviewer’s constructive feedback and insightful questions, which have helped us improve the clarity and completeness of the manuscript. We address each point below:
>
> * How does the proposed lexical mapping handle cases where the grammatical subject does not correspond to the true visual referent (e.g., “A man is walking a dog”)?
>
> Our lexical mapping function relies on automatic subject extraction to identify the primary semantic concept in each generated description. We acknowledge that this mechanism can overlook cases where the true visual referent is mentioned in the sentence but not expressed as the grammatical subject. However, this behavior aligns with the definition of the open-ended segmentation task: the goal is to produce a precise description of the visual entity corresponding to each mask.
>
> Therefore, our evaluation framework penalizes outputs whose extracted subject does not match the visual referent, even if the entity appears elsewhere in the sentence. In the example “A man is walking a dog,” when evaluating a dog mask, a caption whose subject is “man” is considered incorrect because it fails to treat the dog (the true visual referent) as the central entity being described. In contrast, if the model predicts “A dog being walked by a man,” our evaluation framework will consider it accurate.
>
> In short, the lexical mapping does not attempt to reinterpret or reorder linguistic structures; instead, it enforces that generated descriptions explicitly foreground the correct visual entity, aligning with the fundamental objective of open-ended segmentation: detailed and accurate, instance-level recognition of the objects present in the image.
>
> * Does the method explicitly account for modifiers such as adjectives or compound nouns (e.g., “toy dog,” “hot dog,” “drawing of a dog”)?
>
> Our approach can recognize compound nouns in generated descriptions by including them in the initial set of candidate nouns and explicitly considering them in our automatic subject extraction step. However, current standard segmentation datasets typically lack attribute-level annotations, meaning they do not provide ground-truth labels for modifiers, such as adjectives, of visual entities within an image. As a result, our framework cannot fully evaluate the correctness of modifiers because the required attribute annotations are unavailable. We will include the clarification regarding accounting for compound nouns in the “Evaluation Metrics” subsection and the consideration for modifiers in the final version.
>
> * Several closely related works [1,2,3] are missing from the discussion.
>
> We thank the reviewer for pointing out these references. We have updated the “Related Work” section in the revised paper to describe their methodological approach. Specifically, these state-of-the-art works have gone beyond open-vocabulary segmentation and incorporated language models to automatically generate the input vocabulary solely from visual information, to finally perform mask classification.

---

> ### Author Response · Authors · 2025-11-21
>
> * Since the lexical mapping depends on large language models for synonym, hyponym, and meronym extraction, how do the authors address potential biases or inconsistencies in these LLM-generated resources?
>
> To address the current bottleneck in evaluation metrics for open-ended visual segmentation, our framework leverages the broad world knowledge and generalization capabilities of LLMs while maintaining an interpretable evaluation pipeline. We agree with the reviewer that, since our lexical mapping relies on LLMs to extract synonyms, hyponyms, and meronyms, there may still be biases or inconsistencies associated with the LLM. However, our human verification results confirm that the semantic relationships captured by the mapping align well with human judgment, which enables an initial assessment of how sensitive our framework is to potential LLM-generated inconsistencies. Our empirical results suggest that our design choice of leveraging lexical relationships to make the evaluation more interpretable mitigates the biases introduced by relying on pretrained models.
>
> Moreover, to further assess robustness to LLM-specific variability, we performed an ablation in which the lexical mapping process was repeated using two different LLMs. For all ablation experiments, we report the median results for OPAL and Osprey to account for computational complexity. To ensure fairness, we used identical prompts and set the temperature to 0.1 for all LLMs, minimizing stochasticity. Table 2 presents open-ended segmentation results for panoptic (PQ), semantic (mIoU), and instance (AP) segmentation tasks, evaluated under our proposed evaluation protocol on the validation sets of ADE20K and Cityscapes, using GPT-4 and Gemini lexical mappings. The observed performance differences are negligible, and method ranking is preserved, providing additional empirical evidence that our framework is not overly sensitive to any particular LLM’s biases. We include these results in the revised version.
>
> Table 2. **Lexical mapping impact across different LLMs**. We present open-ended segmentation results to compare the effect of using two different LLMs for the lexical mapping process. We include open-ended segmentation results for panoptic (PQ), semantic (mIoU), and instance (AP) segmentation tasks, evaluated using our proposed evaluation protocol on the validation sets of the ADE20K and Cityscapes datasets.
>
> | |    |            |      **ADE20k**    |       |           |   **Cityscapes**    |       |
> |--------|--------|----------------------|-------|-------|----------------------|-------|-------|
> |   **Method**     |  **LLM**      | PQ                   | mIoU  | AP    | PQ                   | mIoU  | AP    |
> | Opal   | GPT-4  | 47.9                 | 37.9  | 37.6  | 52.4                 | 55.3  | 32.7  |
> |        | Gemini | 48.0                 | 37.7  | 37.1  | 48.3                 | 51.7  | 30.2  |
> | Osprey | GPT-4  | 45.6                 | 36.4  | 35.5  | 49.0                 | 53.7  | 30.1  |
> |        | Gemini | 45.6                 | 36.2  | 35.0  | 46.2                 | 51.7  | 28.2  |

---

> ### Author Response · Authors · 2025-11-21
>
> * Could the authors discuss how their framework might generalize to fine-grained or domain-specific datasets such as LVIS or COCO-Stuff?
>
> To ensure compatibility and fairness with state-of-the-art methods (e.g., Osprey, Ferret, MasQCLIP), we adhere to the standard experimental setup for open-vocabulary segmentation evaluation. In this setup, to approximate an open scenario, models are evaluated solely on datasets not included in the training data, typically excluding the MS-COCO dataset used by all state-of-the-art methods. Since the LVIS dataset provides annotations only for MS-COCO training images, and the COCO-Stuff dataset is also included in the training data, they do not align with this experimental setup and would not allow us to benchmark the open-ended segmentation task.
>
> * Given that the experiments assume pre-defined masks as input, does the proposed setup still align with the notion of “open-ended segmentation”?
>
> Our evaluation protocol and experimental setup are aligned with the open-ended segmentation task, which requires generating both meaningful segments and accurate language descriptions. In response to the reviewer’s feedback, we have clarified this definition in the “Task Formulation” subsection on the revised version of our paper. Specifically, our lexical alignment metric jointly assesses pixel-level localization and semantic recognition of visual entities by leveraging established segmentation metrics. Additionally, we evaluate semantic generalization by training on one dataset and testing on another with only partial vocabulary overlap, following standard practice in open-vocabulary segmentation.
>
> Regarding our method, prior state-of-the-art models (e.g., Osprey) in this area have decoupled localization from recognition, treating segmentation masks as given inputs. By decomposing the task into two subtasks (i.e., mask localization and open-ended visual recognition), advances in either area can potentially benefit the other. Following this approach, OPAL advances open-ended segmentation by focusing on generating accurate language descriptions of visual entities. This separation allows focused progress on the recognition component independently of localization performance.

---

> > ### Comment · Reviewer_zkjh · 2025-11-26
> >
> > I thank the authors for their careful response, and I will maintain my positive rating.

---

> > > ### Author Response · Authors · 2025-11-27
> > >
> > > Thank you for your kind response and for maintaining your positive rating. We are glad our rebuttal addressed your questions, and we appreciate your thoughtful feedback. Your comments helped us improve our work, and we have included all suggested revisions in the final version of our paper.

---

### Author Response · Authors · 2025-12-03
**Rebuttal Discussion Summary**

We thank the AC for their effort in evaluating our work. We now summarize the rebuttal discussion, which successfully addressed all reviewers' questions and further supports the unanimous positive assessment of our work.

We thank the reviewers for their valuable feedback and for recognizing the contributions of our work. The reviewers highlighted the novelty of our lexical relationship–aware mapping and the Lexical Alignment Curve (LAC), noting that it provides a systematic, interpretable, and strong human-aligned framework for evaluating free-form outputs in open-ended segmentation (zkjh, BTMF, 5AQi). They emphasized the conceptual elegance and clarity of our formulation, and its significance in addressing a key methodological gap in open-ended visual understanding (zkjh, BTMF). Reviewers acknowledged the rigor of our empirical analysis, including human studies, metric development, and comprehensive re-benchmarking. They also recognized OPAL as the first contrastively trained multimodal LLM for this task, achieving state-of-the-art performance on ADE20K and Cityscapes (zkjh, 5AQi). We appreciate these insights and the constructive feedback that helped us further strengthen our work. Below, we summarize the points clarified during the rebuttal phase:

**zkjh:** We clarified that our lexical mapping intentionally treats incorrect subject–referent alignment as an error, since open-ended segmentation requires generated descriptions to explicitly foreground the visual entity corresponding to each mask. Additionally, we explained that compound nouns are supported, but our framework cannot fully evaluate modifier correctness because the required attribute annotations are unavailable; we will clarify this in the final version. We have updated the “Related Work” section in the revised paper to describe [1-3] methodological approach. To address concerns about LLM-derived lexical relations, we reported an ablation study using multiple LLMs. This study showed negligible variance and preserved method ranking, supporting robustness. Furthermore, we clarified why LVIS and COCO-Stuff cannot be included under the standard open-ended evaluation protocol, due to training data overlap. Finally, we explained that using pre-defined masks aligns with prior open-ended segmentation work that decouples localization from recognition. The reviewer found our clarifications sufficient and kept a positive score.

**BTMF:** We clarified that by decoupling the evaluation protocol from the model and leveraging a standard implementation, we ensure that our work not only enables fair comparisons but also facilitates future research and reproducibility. Moreover, we addressed concerns about reliance on external LLMs by providing human-verification results and an ablation across multiple LLMs, showing negligible differences and preserved ranking. Additionally, we explained why the LAC metric cannot be hacked by hyponyms or meronyms, since the area-under-curve formulation heavily prioritizes early (exact/synonym) matches. We also clarified how our mapping handles polysemy and contextual semantics, and how the framework can naturally extend to open-ended detection. All resources, including the pre-computed lexical mapping, are included in the Supplementary Material to support reproducibility. The reviewer found the rebuttal satisfactory and had no further questions.

**5AQi:** We clarified that, following the standard open-ended evaluation protocol used in prior work, datasets such as COCO, LVIS, and COCO-Stuff cannot be used for testing because they overlap with the training data of all SOTA methods. We also added ablations comparing  multiple LLMs for lexical mapping, showing negligible differences and strengthening reproducibility. Further, we included ablation experiments to assess the effects of noun list size on mapping performance. Our findings confirm that the nouns in our mapping reliably represent the semantic concepts generated by an LLM in open-ended recognition tasks. Finally, we describe how our approach achieves a good trade-off between computational complexity and accuracy. The reviewer found our clarifications satisfactory and continues to support acceptance.

---

### Meta-Review · Area_Chair_6rh8 · 2026-01-11

**Summary:**

This submission targets a real pain point in open-ended segmentation evaluation: embedding-based metrics can disagree with human judgments. The paper proposes a lexical relationship–aware mapping and aggregates it into the Lexical Alignment Curve (LAC) to produce a more interpretable, human-aligned evaluation protocol. They then re-benchmark prior methods under this protocol and introduce OPAL, a contrastively trained multimodal LLM for region-to-text alignment, reporting SOTA on ADE20K/Cityscapes under the proposed evaluation. Overall, the reviewer panel is uniformly above-threshold with moderate confidence (3–4), and post-rebuttal sentiment is clearly positive.

**Reviewer Concerns:**

**Addressed by rebuttal:**
- Subject–referent mismatch (zkjh): The authors clarified this as an intentional penalty aligned with the task objective, which the reviewer accepted.
- Modifiers / compound nouns (zkjh): They clarified compound nouns are supported, but modifier correctness cannot be fully evaluated due to lack of attribute annotations; they commit to clarifying this limitation in the final.
- LLM-derived lexical relations / reproducibility / bias (zkjh, BTMF, 5AQi): Added cross-LLM ablation (GPT-4 vs Gemini) showing negligible differences and preserved ranking; also promise release of resources and provide precomputed mappings in supplementary.
- “Metric can be hacked” (BTMF): Authors clarified the AUC-over-levels design and showed improved human alignment; the reviewer was satisfied.
- Vocabulary coverage (5AQi): Added a noun-list-size ablation (20–100%) showing monotonic gains and stable rankings. Related work: Updated to include the cited papers.
- Related work omissions (zkjh): Authors updated related work to include the cited papers.

**Remaining concerns (but not acceptance-blocking):**
- Generalization to additional datasets: Still limited to ADE20K/Cityscapes; excluding COCO/LVIS/COCO-Stuff is reasonable for comparability but narrows external validity.
- Task framing ambiguity: Masks are assumed given, but the title/framing may over-claim unless stated explicitly.
- OPAL efficiency reporting: Compute is partially addressed (≈1% training FLOPs; same inference), but lacks full latency/memory results— a minor weakness given the paper’s benchmark focus.

**Reviewer Scores:**

- zkjh: 6, confidence 4. After rebuttal: explicitly maintains positive rating.
- BTMF: 6, confidence 3. After rebuttal: “no more questions.”
- 5AQi: 6, confidence 3. After rebuttal: states paper is worthy of acceptance.

---

### Decision · Program_Chairs · 2026-01-26

Accept (Poster)